# Computational Design of Gas Sensors Based on V_3_S_4_ Monolayer

**DOI:** 10.3390/nano12050774

**Published:** 2022-02-25

**Authors:** Ilya V. Chepkasov, Ekaterina V. Sukhanova, Alexander G. Kvashnin, Hayk A. Zakaryan, Misha A. Aghamalyan, Yevgeni Sh. Mamasakhlisov, Anton M. Manakhov, Zakhar I. Popov, Dmitry G. Kvashnin

**Affiliations:** 1Center for Energy Science and Technology, Skolkovo Institute of Science and Technology, Bolshoy Boulevard 30, bld. 1, 121205 Moscow, Russia; i.chepkasov@skoltech.ru; 2Emanuel Institute of Biochemical Physics RAS, 4 Kosygin Street, 119334 Moscow, Russia; yekaterina.sukhanova@phystech.edu (E.V.S.); zipcool@bk.ru (Z.I.P.); dgkvashnin@phystech.edu (D.G.K.); 3Computational Materials Science Laboratory at the Center of Semiconductor Devices and Nanotechnology, Yerevan State University, 1 Alex Manoogian St., Yerevan 0025, Armenia; hayk.zakaryan@ysu.am (H.A.Z.); misha.aghamalyan@gmail.com (M.A.A.); 4Department of Molecular Physics, Yerevan State University, 1 Alex Manoogian St., Yerevan 0025, Armenia; y.mamasakhlisov@ysu.am; 5Department of Materials Technology and Structure of Electronic Technique, Russian-Armenian University, 123 Hovsep Emin St., Yerevan 0051, Armenia; 6Aramco Innovations LLC, Aramco Research Center, 119234 Moscow, Russia; anton.manakhov@aramcoinnovations.com

**Keywords:** 2D, nanomaterials, vanadium chalcogenides, monolayer, gas sensor, dft

## Abstract

Novel magnetic gas sensors are characterized by extremely high efficiency and low energy consumption, therefore, a search for a two-dimensional material suitable for room temperature magnetic gas sensors is a critical task for modern materials scientists. Here, we computationally discovered a novel ultrathin two-dimensional antiferromagnet V_3_S_4_, which, in addition to stability and remarkable electronic properties, demonstrates a great potential to be applied in magnetic gas sensing devices. Quantum-mechanical calculations within the DFT + *U* approach show the antiferromagnetic ground state of V_3_S_4_, which exhibits semiconducting electronic properties with a band gap of 0.36 eV. A study of electronic and magnetic response to the adsorption of various gas agents showed pronounced changes in properties with respect to the adsorption of NH_3_, NO_2_, O_2_, and NO molecules on the surface. The calculated energies of adsorption of these molecules were −1.25, −0.91, −0.59, and −0.93 eV, respectively. Obtained results showed the prospective for V_3_S_4_ to be used as effective sensing materials to detect NO_2_ and NO, for their capture, and for catalytic applications in which it is required to lower the dissociation energy of O_2_, for example, in oxygen reduction reactions. The sensing and reducing of NO_2_ and NO have great importance for improving environmental protection and sustainable development.

## 1. Introduction

The gas-sensing system is important in various applications including industrial pollutant gas leakage detection, environmental monitoring, medical care, food industry, etc. [1]. Nanomaterials show great performance in this field of technology [2]. Nowadays, the most effective materials for gas sensing belong to the groups of metal-organic framework-based nanostructured materials [3,4] and low-dimensional materials [5,6]. Another type of sensor is based on magnonic sensors with the presence of magnetic nanoparticles [7] and two-dimensional materials [1,8] and it seems to be promising for further development.

Magnetic gas sensors are characterized by greater safety of use, lower working temperatures, and faster response times than traditional gas sensors based on the variation of electronic properties. The change in the magnetic properties of the active material can be detected, for example, by the Hall effect observation, change in magnetization or electron spin orientation, ferromagnetic resonance modification, magneto-optical Kerr effect, and magneto-static wave oscillation effect [9,10].

Two-dimensional materials that belong to the family of transition metal dichalcogenides (TMD) have attracted great interest in the field of gas sensors due to a large surface-to-volume ratio, leading to specific electrical properties. These properties can be strongly modified by the surface adsorbates [11], which is confirmed by recent theoretical studies (MoS_2_ [12], SnS_2_ [13], and VS_2_ [14]). However, there is still a lack of critical information about the real possibility of using TMD monolayers as a magnetic gas sensor component. Most known TMDs are non-magnetic [15,16], but it was shown [17,18,19,20,21,22] that adding point defects or adsorption of non-metallic elements can make them magnetic.

Among the TMD family, the VS_2_ monolayer has attracted particular interest due to its intrinsic magnetism [23,24,25] which can be tuned by, for example, tensile strain [26]. Ferromagnetism in the VS_2_ layer was experimentally confirmed at room temperature by Gao et al. [23] where an unusual hybridization of 3*d* orbitals of V and 3*p* orbitals of S was also observed. Zhong et al. [27] determined that Curie temperature (Tc) strongly depends on the thickness, and the Tc of single-layer VS_2_ was determined to be 72 K.

Despite the extensive study of VS_2_, there is still an open question related to the study of 2D monolayers in the V–S system with compositions differing from VS_2_. Recent studies have shown that in M–X systems (M = W, Mo, etc.; X = S, Se, Te, etc.), new two-dimensional and non-stoichiometric structures can be formed [28]. There are several studies devoted to vanadium sulfide crystals such as VS_4_ [29,30], V_3_S_4_ [31,32], V_5_S_4_ [33], V_5_S_8_ [34], and V_3_S [35] structures in different applications. For example, the bulk V_3_S_4_ usually forms a distorted NiAs-type structure with single-layer VS_2_ building blocks and additional V atoms bonded between two layers with metallic characteristics [36]. Since the properties of two-dimensional materials differ from the bulk ones, this significantly expands the scope of their application. Motivated by the above-mentioned facts, we performed a complex investigation toward designing new two-dimensional V–S phases suitable for magnetic gas sensors. Among the considered earlier structures, we predicted a novel magnetic V_3_S_4_ monolayer demonstrating stability as promising for applications as an element of CO and NO molecule magnetic gas sensors.

## 2. Materials and Methods

The global search for thermodynamically stable 2D compounds in the V–S system was performed by using the variable-composition evolutionary crystal structure prediction algorithm USPEX [37,38,39]. The first 180 structures (first generation) were generated by the operator of random symmetry [39] in the unit cell with up to 16 atoms in the primitive cell. Further generations (120 structures in each) consisted of 20% of the structures of random symmetry [39] and 80% of the structures were generated by the operators of heredity, soft mode mutation, and transmutation.

For each considered compound, the structure was optimized using density functional theory (DFT) [40,41]. The generalized gradient approximation (GGA) was used with the Perdew–Burke–Ernzerhof (PBE) parameterization for the exchange-correlation functional [42] as implemented in the Vienna Ab initio Simulation Package (VASP) 6.1.2 [43,44,45,46]. Ion-electron interaction was described by the augmented plane waves method (PAW) [47], and the cutoff energy of plane waves was set to 500 eV. The partition of the first Brillouin zone into a grid of k-points was carried out within the Monkhorst–Pack scheme [48] with a resolution of $2\pi \times 0.05\;{Å}^{-1}$. To consider strong electron correlations between the localized 3*d*-electrons of V atoms, the GGA + *U* approach in Dudarev’s formulation [49,50] was applied, where the values of parameter *U_eff_*
*= U* − *J* varied from 0 to 4 eV. The Grimme corrections (DFT-D3) [51] were applied to take into account van der Waals interactions between the V_3_S_4_ surface and molecules.

The phonon density of states for V_3_S_4_ was calculated using the finite displacements method as implemented in the PHONOPY program package (version 2.11.0) [52,53] with forces computed by VASP [43,44,45,46]. For phonon calculation, the 4 × 2 × 1 supercell was used. Phonon dispersion curves were plotted using sumo software (version 2.2.5) [54].

## 3. Results and Discussion

### 3.1. Crystal Structure Prediction

Two-dimensional compounds in the V–S system were predicted by using the variable-composition evolutionary search as implemented in USPEX [37,38,39] code. The V–S system belongs to strongly correlated systems. Consequently, it is necessary to use the GGA + *U* approach to correctly describe the stability and electronic properties of V–S compounds, as suggested in [55,56]. As we considered many different compositions and structure types in a single evolutionary search, we could not use one specific *U_eff_* value. Thus, we performed five independent evolutionary searches with different *U_eff_* values from 0 to 4 eV with the step of 1 eV and the resulting convex hulls are summarized in Figure 1. Besides pure compounds, we predicted four thermodynamically stable monolayers with various compositions, namely, V_3_S, VS, V_3_S_4_, and VS_2_ (see Appendix A). In the resulting convex hull H, polymorphic modification of VS_2_ becomes less energetically favorable than the T phase at the *U_eff_* value >2 eV, which is consistent with a previous study by Zhuang et al. [55]. At the same time, VS_2_ and VS composition disappeared from the convex hull at the *U_eff_* value equal to 4 eV and 3 eV, respectively. V_3_S_4_ and V_3_S phases were thermodynamically stable at all considered *U_eff_* parameter values. For all further calculations, we finally employed the value of *U_eff_* = 3 eV [55].

### 3.2. Novel V_3_S_4_ Phase

We predicted a novel monolayer structure of V_3_S_4_ that was thermodynamically stable at all considered *U_eff_* values (see Figure 1 and Appendix A). This result was proved by an additional fixed-composition evolutionary search of V_3_S_4_ that showed that the obtained structure had the lowest formation energy among other structures with the same composition. Detailed information is presented in the Appendix A. Top and side views of the atomic structure of the novel V_3_S_4_ phase are shown in Figure 2a. The V_3_S_4_ phase has the cell parameters of *a* = 3.35 Å, *b* = 6.75 Å, and *γ* = 104.35°. The unit cell of the V_3_S_4_ monolayer consists of three vanadium atoms, among which one atom exists in octahedral surroundings of sulfur atoms (marked by blue in Figure 2a), and two other vanadium atoms exist in a square pyramidal surrounding (C_4v_) as marked by green in Figure 2a. Suddenly, one can find analogies of V_3_S_4_ monolayer in bulk crystals [57] in which alternation of structural elements of square pyramidal and octahedrally coordinated vanadium atoms are presented. Moreover, the exfoliation of the V_3_S_4_ crystal from [33] along the $\left[20\bar{1}\right]$ direction could give a monolayer with the same structure as predicted in our USPEX calculations. Thus, it can be assumed that the presence of a two-dimensional monolayer counterpart of the bulk crystals provides prerequisites for their experimental synthesis.

The dynamical stability of the predicted V_3_S_4_ monolayer was studied by calculations of the phonon band structure as shown in Figure 2b. There are minor imaginary modes in the vicinity of the Γ-point in the obtained spectra usually associated with out-of-plane vibrational modes caused by the small supercell size, and therefore, we can conclude the dynamic stability of the predicted novel structure.

To examine in detail the electronic and magnetic properties of the V_3_S_4_ phase, we first determined the monolayer’s magnetic ground state. For this aim, we considered the 2 × 1 × 1 supercell of the V_3_S_4_ phase in the ferromagnetic (FM) and 10 different antiferromagnetic (AFM) configurations (for more details, see Appendix A). The comparison of the total energies of these magnetic configurations is shown in Appendix A. The most energetically favorable configuration was noted as AFM10 and the energy difference between the FM and AFM10 configuration was 0.15 eV per unit cell, which is relatively large, indicating the stability of antiferromagnetic configuration. The electronic band structure for the ground state calculation is presented in Figure 2c. Vanadium atoms in octahedral surroundings have a magnetic moment equal to 1.97 μB, while vanadium atoms in the square pyramidal surrounding have a magnetic moment of 2.14 μB. The novel V_3_S_4_ monolayer in the AFM magnetic state exhibited semiconducting properties with the band gap of 0.36 eV.

The main feature of V_3_S_4_ is the existence of two symmetrically inequivalent vanadium atoms. Furthermore, and more importantly, vanadium atoms in square pyramidal surroundings have a vacant *d*-orbital that can form a bond with different ligands; for example, with gas molecules, and depending on the binding characteristic, it should lead to a change in cleavage after binding to a gas molecule. This will affect the change in the electronic properties of V_3_S_4_, thus, the surface of the V_3_S_4_ monolayer is promising as a gas sensor with a large number of adsorption centers.

### 3.3. Sensing Properties

A study of V_3_S_4_ as a potential gas sensor was performed for the cases of the adsorption of CO, CO_2_, NO, NO_2_, NH_3_, H_2_O, and O_2_ molecules. For this study, the 4 × 2 × 1 supercell of V_3_S_4_ was considered containing 24 vanadium and 32 sulfur atoms. For each molecule, a various number of possible adsorption sites were considered. The most energetically favorable sites for all molecules correspond to the vanadium atom of the outer layer in the square pyramidal surrounding (see Appendix A).

Detailed information on the adsorption of different gas molecules on V_3_S_4_ nanosheet is presented in Table 1. The energy of adsorption (*E_a_*) was calculated as *E_a_* = *E[V_3_S_4_* + *molecule]* − *E[V_3_S_4_]* − *E[molecule]*, where *E[V_3_S_4_* + *molecule]* is the energy of the system consisting of the interacting V_3_S_4_ monolayer and gas molecule, and *E[V_3_S_4_]* and *E[molecule]* are the total energies of the V_3_S_4_ nanosheet and pristine molecule, respectively. Considering the sign of the charge transferred to the V_3_S_4_ monolayer, we can conclude that CO, CO_2_, NO, NO_2_, and O_2_ molecules act as p-type dopants, while NH_3_ and H_2_O molecules act as n-type dopants, as indicated by the positive sign of Δ*q* (see Table 1). In systems with strong charge overflow such as in the cases of O_2_, NO, and NO_2_ molecules, the sorption of extra electrons at the molecule leads to the occupation of antibonding orbitals increasing the molecule bond length (see Table 1). High values of adsorption energies, except for the CO_2_ molecule, are related to the pronounced charge redistribution in the interface area. However, in the case of the CO_2_ molecule, the obtained values of *E_a_* and *D* and nearly zero charge redistribution suggest weak van der Waals interactions between the molecule and monolayer.

During the adsorption of molecules, except for O_2_ and NO_2_, the surrounding of the vanadium atom changes from a square pyramid to an octahedral one. In the case of absorbed oxygen or nitrogen dioxide molecules, the vanadium atom formed the capped trigonal prismatic coordination because both oxygen or nitrogen and oxygen atoms interact with vanadium atoms.

For a detailed analysis of the change in the electronic configuration of the sorption site of the vanadium atom, we analyzed the density of electronic states (Figure 3) and electronic band structures (Appendix A). Despite the relatively high binding energy between the molecule and the substrate, molecule sorption does not always lead to a significant change in magnetic and electronic properties, as exemplified by the cases of CO, NH_3_, and H_2_O molecules sorption because their HOMO and LUMO states were located quite far away from the band gap of the V_3_S_4_ nanosheet. Therefore, the V_3_S_4_ monolayer is not appropriate for application as a detector of CO, CO_2_, NH_3_, and H_2_O molecules.

It is noteworthy that the HOMO and LUMO states of NO_2_ molecules are also located quite far from the band gap of V_3_S_4_, however, its sorption leads to the emergence of an unoccupied electronic state in the band gap of V_3_S_4_ related mostly with vanadium atoms, which leads to a band gap reduction to 0.14 eV. At the sorbed configuration of NO_2_ molecules on the V_3_S_4_ surface, the geometry of the molecule dramatically changed due to the electron donation from V_3_S_4_ with filling of the antibonding orbital. This led to elongation of one of the N–O bonds to 1.42 Å. Such a feature indicates not only sensoric applications of V_3_S_4_, but can also be useful for the catalytic reduction of NO_2_ [60].

In contrast, at the adsorption of O_2_ and NO molecules, some electronic levels in the band gap were observed. The sorption of the oxygen molecules led to the appearance of unoccupied electronic states above the valence band maximum for one spin channel and band gap reduction to 0.24 eV. The molecule attracts electron density (0.473 *e*) due to high electronegativity to an antibonding orbital, which leads to an increase in O–O distance by 0.09 Å in comparison with the pure O_2_ molecule. This allows one to conclude about the viability of V_3_S_4_ in catalytic applications in which it is required to lower the dissociation energy of molecular oxygen, for example, in oxygen reduction reactions [61,62].

In the case of NO molecule sorption, occupied states close to the valence band maximum and unoccupied states close to the Fermi level for one spin channel were observed, leading to the band gap reduction of 0.14 eV. The magnetic moment of the vanadium atom in a square pyramidal surrounding located under adsorbed NO molecule changes from 2.14 μ_B_ to 1.67 μ_B_. Taking into account that the electric conductivity *σ* (which is the main parameter determining the sensitivity of the sensor) is proportional to *σ~exp(-E_g_/k_B_T)* (where *E_g_* is the band gap, *k_B_* is the Boltzmann’s constant, and *T* is the temperature), we can conclude that due to significant band gap reduction caused by the sorption of NO and NO_2_ molecules, the V_3_S_4_ monolayer is a prospective material for NO and NO_2_ molecule detection. The detection of nitric oxide and nitric dioxide is of great importance for environmental protection as these harmful gases are emitted from the upstream of oil and gas production [63]. The development of novel materials that can be used both as sensors and catalysts for NO and NO_2_ reduction will positively affect sustainable development.

## 4. Conclusions

In this work, we performed a variable-composition evolutionary search for new stable two-dimensional V–S monolayers with different *U_eff_* values (from 0 to 4 eV) via the USPEX algorithm. The new 2D V_3_S_4_ structure was found to be thermodynamically and dynamically stable, showing intriguing electronic and magnetic properties for sensing applications. We found that the V_3_S_4_ monolayer exists in an antiferromagnetic ground state and shows semiconducting behavior with a band gap of 0.36 eV. These features signal that the V_3_S_4_ monolayer can be used as a gas sensor for different gas agents. We proved this idea by calculating the energies of adsorption and electronic properties of V_3_S_4_ with adsorbed CO, CO_2_, NO, NO_2_, NH_3_, H_2_O, and O_2_ molecules. Detailed analysis of the electronic properties of V_3_S_4_ showed its prospective to be used as effective sensing materials for NO_2_ and NO gas molecules and as a material for catalytic applications in which it is required to lower the dissociation energy of O_2_, for example, in the oxygen reduction reactions. The obtained results pave the way for using transition metal chalcogenide monolayers in gas sensors.

## Figures and Tables

**Figure 1 nanomaterials-12-00774-f001:**
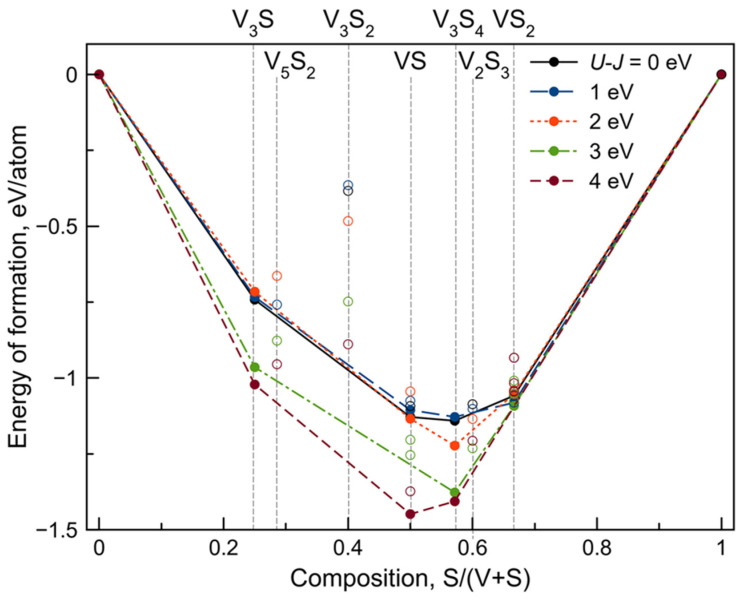
Calculated convex hulls of the V–S system with different *U_eff_* values.

**Figure 2 nanomaterials-12-00774-f002:**
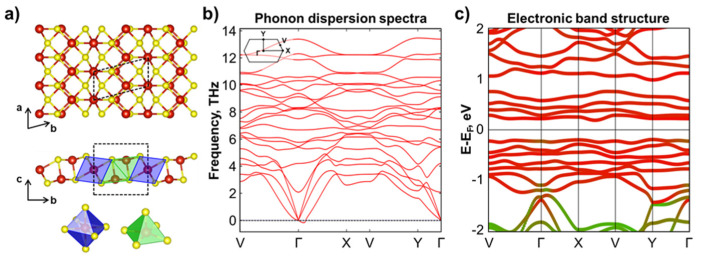
Newly predicted V_3_S_4_ monolayer: (**a**) top and side views of the atomic structure (color legend: V—red; S—yellow); (**b**) phonon band structure; (**c**) electronic band structure of the most energetically favorable AFM configuration. The contributions from vanadium and sulfur atoms are indicated with red and green colors, respectively.

**Figure 3 nanomaterials-12-00774-f003:**
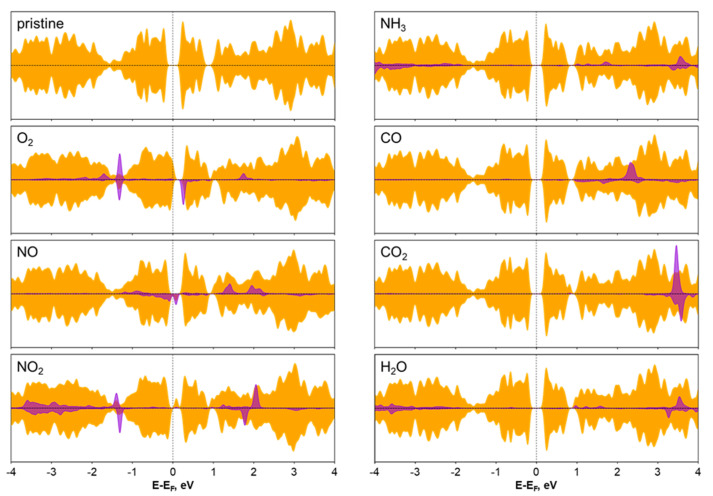
The density of electronic states of the V_3_S_4_ monolayer before and after the adsorption of CO, CO_2_, H_2_O, O_2_, NH_3_, NO, and NO_2_ gas molecules. Orange and violet colors denote the contribution from the V_3_S_4_ monolayer and gas molecules, respectively. The Fermi level shifted to zero. The contribution from gas molecules was enlarged by 4 times for CO, CO_2_, O_2_, NO, and NO_2_ molecules and by 20 times for the H_2_O and NH_3_ molecules.

**Table 1 nanomaterials-12-00774-t001:** Characteristics of the most energetically favorable adsorption sites in the V_3_S_4_ nanosheet with different molecules adsorbed to the surface. *E_a_* is the energy of adsorption; *D* is the perpendicular distance between molecule and adsorption site (V atom); α_0_ and α are the bond angles in V-shaped molecules before and after adsorption; *d_0_* and *d* are the molecules bond lengths before and after adsorption; and Δ*q* is the change in charge of the molecule between states before and after adsorption (within the Bader theory calculation [58,59]).

Molecule	*E_a_* (*e*V)	*D* (Å)	α_0_ (°)	α (°)	*d_0_* (Å)	*d* (Å)	Δ*q* (*e*)
CO	−0.82	2.11	-	-	1.14	1.14	0.069
CO_2_	−0.26	2.39	179.97	179.48	1.17	1.17	0.007
H_2_O	−0.83	2.23	104.35	106.48	0.97	0.97	−0.051
O_2_	−0.59	2.04	-	-	1.23	1.32	0.473
NH_3_	−1.25	2.22	106.58/106.57/106.56	108.49/108.63/107.59	1.02	1.02	−0.114
NO	−0.93	1.87	-	-	1.16	1.17	0.255
NO_2_	−0.91	1.91	133.9	111.37	1.21	1.42/1.19	0.506

## Data Availability

Not applicable.

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
