# Peer review of "Computational Design of Gas Sensors Based on V3S4 Monolayer"

_nanomaterials, 2022, doi:10.3390/nano12050774_

Round 1
Reviewer 1 Report
The manuscript give a new material design and calculation for V3S4 magnetic gas sensors. The result is of some novelty and reasonability, but the calculation is too simple to get more valuable result, for example, site of adsorption, mode of adsorption, band gap, electron population, electron tranfer and so on. In addition, V3S4 has been reported its application in catalyst and stored energy, so it is not a new materials. As your study, magneteic gas sensor is not precise rather than electrical resistance.
Author Response
Q1. The manuscript give a new material design and calculation for V3S4 magnetic gas sensors. The result is of some novelty and reasonability, but the calculation is too simple to get more valuable result, for example, site of adsorption, mode of adsorption, band gap, electron population, electron tranfer and so on. In addition, V3S4 has been reported its application in catalyst and stored energy, so it is not a new materials. As your study, magneteic gas sensor is not precise rather than electrical resistance.
A1. We thank the reviewer for the comments. We agree with the referee that V3S4 was investigated previously in the form of bulk material, but in our investigation, we have revealed the new form having such stoichiometry – a two-dimensional monolayer was predicted unbiasedly via the USPEX algorithm. Since the properties of two-dimensional materials differ from the bulk ones, this significantly expands the scope of their application. For example, based on electronic structure calculation the bulk form of V3S4 has the metallic character, on the contrary, the monolayer form has semiconducting properties which extend the scope of V3S4 in the two-dimensional form to the sensor area. In addition, monolayers have a larger surface area, which also makes them more suitable for sensor applications.
Compared to their counterparts - electrical property-based gas sensors[1], magnetic gas sensors have emerged as a more attractive candidate due to the following reasons. (i) No electrical contacts are needed to detect the gas, which lowers the risk of explosion due to fire when used in hydrogen-powered vehicles or in the presence of reactive chemicals or pollutants, (ii) the magnetic response is much faster compared to chemiresistive sensors.
(1) Shinde, P. V.; Rout, C. S. Magnetic Gas Sensing: Working Principles and Recent Developments. Nanoscale Advances 2021, 3 (6), 1551–1568.
Reviewer 2 Report
The Authors report a computational study on the potential of ultra-thin two-dimensional antiferromagnetic V3S4 for the development of gas sensors based on magnetic material. The authors studied the electronic and magnetic properties of V3S4 in the presence of various gases.
The document is well presented and is ready for publication after minor revisions.
-The authors should explain how the study could be influenced by a different behavior when the gas is absorbed in the "bulk" and not only in the surface (in the case of a non monolayer structure).
-The authors could add some comments on the possible effects of the presence of structural defects, as always happens in the real case.
Reviewer 3 Report
The manuscript entitled “Computational Design of Gas Sensors Based on V3S4 Monolayer” This work is of great interest to the gas sensor community. Nonetheless, the manuscript can be further improved by considering the following aspects.
Main comments
- Introduction: Please justify how the work is novel since the Vanadium sulfide-based gas sensing application has already been reported.
- Abstract: important information is missing, such as critical results
- The first paragraph contains trivial statements. The introduction should be reduced in length and focus on current analytical challenges. Essential related works can be cited.
- In the abstract, it is important to mention significant findings
- The quality of some figures is inferior and needs to be enhanced.
- In the introduction, it is better to include some strategies to improve selectivity and sensitivity at room temperature. Take a look and cite relevant references: Some suggested review articles to cite in the intro. Part: Sensors 2019, 19(2), 233, Adv. Mater. 2016, 28, 795–831, Microchimica Acta, 185(2018) 213, Adv. Funct. Mater. 2017, 27, 1702168.
- It is better to check and correct the font size of the x and y-axis of all figures in the manuscript. It should be the same.
Round 2
Reviewer 1 Report
The revisions can be accepted. So I approve its publication.